# The Effect of Vitamin D Consumption on Pro-Inflammatory Cytokines in Athletes: A Systematic Review of Randomized Controlled Trials

**DOI:** 10.3390/sports12010032

**Published:** 2024-01-13

**Authors:** Saber Saedmocheshi, Ehsan Amiri, Aref Mehdipour, Giuseppe Potrick Stefani

**Affiliations:** 1Department of Physical Education and Sport Sciences, Faculty of Humanities and Social Sciences, University of Kurdistan, Sanandaj 6617715175, Iran; saedsaber384@gmail.com (S.S.);; 2School of Health and Life Sciences, Pontifical Catholic University of Rio Grande do Sul, Porto Alegre 90619-900, Brazil

**Keywords:** vitamin D supplementation, exercise, including interleukin-6, tumor necrosis factor-alpha

## Abstract

Vitamin D is essential for the optimal health of the skeletal system. However, this vitamin also plays a role in other functions of the human body, such as muscle, immune, and inflammatory functions. Some studies have reported that adequate levels of vitamin D improve immune system function by reducing the levels of certain pro-inflammatory cytokines, which can protect against the risk of post-exercise illness. This systematic review aims to investigate the effects of vitamin D supplementation on pro-inflammatory cytokines in athletes. This study was conducted using the Preferred Reporting Items for Systematic Reviews and Meta-Analyses (PRISMA) guidelines. A literature search was conducted in SPORTDiscus, PubMed, ScienceDirect, and Google Scholar up to 1 October 2023. The quality of the articles was evaluated using the Risk of Bias 2 Tool. After searching the databases, a total of 7417 studies were identified, 6 of which met the eligibility criteria, and their outcomes were presented. The six studies included 176 participants. All six studies are randomized control trials, including a total of 176 subjects, primarily men (81%). Regarding the types of athletes, most participants were endurance athletes. Our investigation in this systematic review demonstrated that out of the six studies, only two of them reported significant changes in IL-6 and TNF-α levels after taking high-dose vitamin D. Other studies did not present any significant changes after vitamin D supplementation in athletes with respect to IL-6 and TNF-α levels. Further studies are needed to determine the effectiveness of vitamin D supplementation for athletes as a disease-prone community.

## 1. Introduction

An excessive exercise-induced inflammatory response is thought to be one of the main factors limiting athletic performance [1]. Pro-inflammatory cytokines can disrupt the metabolism of skeletal muscles and other tissues, for example, via insulin signaling. They can also cause inflammation of the central nervous system, which, in addition to general inflammation, leads to impaired movement regulation and loss of coordination [2]. In a recent study, the prevalence of vitamin D deficiency in extreme endurance athletes and the association between delayed physical performance and vitamin D deficiency during regular exercise were observed [3,4]. These physiological responses in skeletal muscles are influenced by exercise-induced mechanisms and possibly influenced by sports nutrition status and limited exposure to sunlight [5,6].

Sequestered vitamin D is important to the understanding of metabolic disease [7]. Vitamin D deficiency (defined as 25(OH) D levels less than 20 ng/dL) is common and is associated with reduced endurance, performance, and muscle strength [8,9]. Vitamin D deficiency is associated with muscle metabolic disorders, including insulin resistance [10,11], and it is associated with mitochondrial dysfunction in young and older adults [12]. Vitamin D lack is very common in obesity without vitamin D supplementation [13,14]. To combat vitamin D deficiency and vitamin D resistance associated with obesity, high-dose vitamin D supplementation protocols are commonly prescribed for obese adults [15]. Vitamin D is produced in the skin when sunlight is absorbed. Therefore, serum vitamin D levels fluctuate seasonally [16].

During the last decade, much effort has been devoted to understanding the role of vitamin D in modulating the immune system [17]. In vitro studies have indicated that vitamin D-deficient mouse models exhibit impaired macrophage function, including chemotaxis, phagocytosis, and cytokine production [18]. Laboratory research, along with animal studies, has demonstrated that vitamin D enhances the production of anti-inflammatory cytokines, such as transforming growth factor beta-1 (TGF-β1) and interleukin-4 (IL-4), while reducing the production of pro-inflammatory cytokines, including interleukin-6 (IL-6), interferon-gamma (IFN-γ), interleukin-2 (IL-2), and tumor necrosis factor-alpha (TNF-α) [17,19,20]. However, in vivo studies have not been entirely conclusive regarding the impact of pro-inflammatory cytokines like TNF-α on cardiac, skeletal, and bone tissue recovery in high concentrations [21].

Some studies show that vitamin D supplementation improves cytokine profiles in patients with chronic diseases, such as congestive heart failure and osteoporosis [22,23]. On the other hand, other studies in healthy people have not clearly shown the relationship between vitamin D status and the concentration of circulating cytokines [24,25]. It should be noted that muscle strength and power in marathon runners are related to vitamin D levels [21].

Vitamin D deficiency increases the risk of developing muscle myopathy and impaired cross-bridge formation, which leads to muscle weakness and fatigue [25,26]. However, the optimal dose of vitamin D and the serum levels required for exercise performance and recovery have been debated [27,28]. Doses of 600–800 IU/day and 1000 IU of vitamin D may not be sufficient for optimal vitamin D levels and may not prevent the reduction in serum 25(OH)D in response to vigorous exercise [27]. There is evidence that shows that dietary supplements with 2000 to 5000 international units of vitamin D per day have a better effect on bone health and skeletal muscle function [27]. Moreover, low levels of vitamin D or vitamin D deficiency have also been shown to be associated with sarcopenia [29]. Vitamin D deficiency, decreased muscular function, and a higher incidence of sarcopenia have all been linked in numerous human studies [30].

This systematic review describes how exercise interacts with inflammatory cytokines. To the best of our knowledge, no systematic review has investigated the effect of vitamin D consumption on pro-inflammatory cytokines in athletes. Therefore, the purpose of this systematic review is to investigate the effect of vitamin D supplementation on IL-6 and TNF-α in athletes, in which only randomized controlled trails were examined.

## 2. Materials and Methods

### 2.1. Search Strategy and Data Extraction

The protocol of this systematic review was designed according to the Preferred Reporting Items for Systematic Reviews and Meta-Analyses (PRISMA) statement [31]. For all PRISMA checklist (main text and absctract, please see Appendix A, respectively). Two authors independently performed the literature search, study selection, and data extraction. The background search was conducted in the SPORTDiscuss, PubMed, Google Scholar, and ScienceDirect electronic databases; the search was limited to English articles; and it was performed with the following keywords: vitamin D supplement or ergocalciferol or cholecalciferol or inflammation or cytokines, pro-inflammatory cytokines, IL-6, TNF-α, and exercise or sports activity or exercise (all disciplines) or athletes. The time period of the present study extended until 1 October 2023. The indexing words and accessible search terms were clustered according to the PICO scheme (population, intervention, comparator, and outcome) to ensure the search was comprehensive and specific. A manual search of the reference sections of the selected articles was also performed to identify other relevant studies. The search strategy is shown in Figure 1.

Two authors (A.M. and E.A.) screened all of the citations retrieved independently in via a 2-step process. The initial step consisted of screening titles and abstracts using online software (Rayyan) [32], and the second step consisted of reviewing full-text articles to confirm study selection using Foxit Reader software. After each stage, discrepancies were resolved via discussion.

### 2.2. Inclusion and Exclusion Criteria

Studies included in this systematic review met the following inclusion criteria: (1) research conducted with human participants and athletes (people who exercise regularly (i.e., three days a week, eight weeks and more, and the duration of training is 60 min and more per session) were included as athletes); (2) articles that were in English; (3) research articles that examined the effect of vitamin D on inflammatory cytokines levels after exercise; (4) research carried out with a control/placebo group; and (5) published articles. Exclusion criteria were as follows: (1) research conducted with animals; (2) non-English articles; (3) systematic reviews or meta-analyses; (4) studies that included other interventions other than vitamin D supplementation; and (5) observational studies or case–control studies. After applying the inclusion and exclusion criteria, the following data were extracted from each study: name of the first author, year of publication, intervention and placebo group characteristics, vitamin D serum levels, supplement dose, duration of supplementation, exercise or training protocol, and supplementation effects on inflammatory and pro-inflammatory cytokines.

### 2.3. Methodological Quality

Cochrane Collaboration’s tool (ROB2) was used to independently assess risk of bias [33] in a 2-step process. This tool considers bias from randomization and blinding (domain 1), deviations in interventions (domain 2), baseline imbalances (domain 2 for parallel group trials), carryover effects (domain 2 for crossover trials), lack of data (domain 3), outcome measurement (domain 4), and bias in reported outcome selection (domain 5). Each domain, as well as the final judgment, was classified as either “low risk of bias”, “some concerns”, or “high risk of bias”. For a study to be classified as “low risk of bias”, it needed to be considered “low risk” in all 5 domains. Studies were classified as “some concerns” when only 1 domain was classified as “some concerns”. If the study had >1 domain assessed as “some concerns”, or ≥1 domain as “high risk”, the overall study classification would be “high risk of bias” [33] (Figure 2).

## 3. Search Results

The literature search yielded a total of 7417 articles. After reviewing the titles, 7381 articles were excluded due to not studying inflammation and levels of IL-6 and TNF-α after exercise, non-exercising subjects, use of supplements other than vitamin D, lack of RCT studies, or because they were systematic reviews. After removing duplicates, 36 articles were selected for screening based on the title and abstract, and 21 articles were excluded due to their not studying the desired variables or insufficient reporting of the results. Then, 15 studies were selected to read the full text, and the final number of studies in this systematic review was 6. A summary of the search process is shown in Figure 1.

## 4. Study Characteristics

Table 1 indicates the characteristics of included studies. In this research, randomized, double-blind, placebo-controlled studies were examined, which amounted to a total of six studies with 176 subjects. In one study, 24 ultramarathon runners were divided into two groups (n = 12 supplemented group; n = 12 placebo group). The vitamin D group took 2000 IU of the supplement daily and this intervention lasted 3 weeks [34]. In another study, 17 male runners were divided into two control groups (n = 9) and a vitamin D group (n = 8). The vitamin D group consumed 10,000 IU daily for 2 weeks [35]. In addition, in another study, which was conducted on female endurance runners for 6 months, the subjects were divided into two equal groups of 10 subjects (placebo and vitamin D supplemented) and they took 1000 IU of vitamin D daily, consumed with low-fat milk [36]. Moreover, in the study of Lewis [37], which was conducted on swimmers and divers, 4000 IU of vitamin D supplement (19 participants) and a placebo (13 participants) were distributed daily. The supplement group was treated with vitamin D for 6 months. In another study, 35 male marathon runners were divided into two control groups (19 subjects) and a supplement group (16 subjects), and the supplement group consumed 150,000 IU of vitamin D supplement 24 h before the intervention [38]. Moreover, in Todd’s [39] study, which was carried out on 42 soccer players who consumed 3000 IU daily, there were 20 athletes in the placebo group and 22 athletes in the supplement group, and they participated in the research for 12 weeks. In addition, four studies used only male participants [34,35,38,39], one study used mixed subjects [37], and one study used only female participants [36].

In one of the studies, participants consisted of semi-professional ultramarathon runners [38]. Additionally, two other studies involved athletes who had undergone regular training for 3 and 7 years, respectively [34,35]. The remaining three studies did not provide information regarding the training status of the subjects. One study, which focused on semi-professional athletes [38], observed a decrease in IL-6 levels following an ultramarathon (vitamin D dosage: 150,000 IU). Furthermore, another study involving athletes trained for 7 years reported a significant decrease in TNF-α levels one hour after the intervention (vitamin D dosage: 2000 IU/day), as well as a decrease in IL-6 levels 24 h after the intervention (vitamin D dosage: 10,000 IU/day) [34]. It is worth noting that only two studies reported a reduction in IL-6 and TNF-α levels, while the remaining four studies did not indicate any significant decline in IL-6 and TNF-α levels. Additionally, the average vitamin D intake across all six studies was found to be 28.33 IU.

## 5. Discussion

To the best of our knowledge, there are a small set of studies that have investigated the effect of vitamin D supplementation on inflammatory cytokines in athletes. We tried to collect and discuss this research in this systematic review. A total of six studies that were eligible for inclusion were included in the study, and they included a total of 176 participants. All these studies were designed in a randomized control trail manner. The findings of this study show conflicting results regarding the effect of vitamin D on IL-6 and TNF-α levels. Out of the six studies, only two [34,38] demonstrated alterations in IL-6 and TNF-α. In contrast, the remaining studies did not exhibit noteworthy modifications in IL-6 and TNF-α levels as a result of vitamin D intervention in athletes.

There are a number of proinflammatory and anti-inflammatory cytokines that are primarily targeted in sports studies [40]. The main source of most of these cytokines are cells of the immune system, such as T cells, macrophages, B cells, monocytes, and natural killer cells, which can induce inflammatory or anti-inflammatory effects through different mechanisms [41]. Although some studies have studied the changes in cytokines due to the use of vitamin D supplements in athletes, there are still contradictions in their results, which we will mention below.

In a study which included 32 male and female college swimmers and divers [37], the subjects were randomly divided into supplement and placebo groups. In this study, the supplement group consumed 4000 IU of vitamin D daily for 6 months. This 6-month intervention did not show any effect on TNF-α, IL-6 and IL-1β concentrations in the supplement group compared to the placebo group. On the other hand, in a study conducted on a group of collegiate athletes and recreational runners, the amount of TNF-α decreased after vitamin D consumption [21]. Lewis [37] indicated that an insignificant relationship was observed between the level of vitamin D and inflammatory cytokines. On the other hand, in a study conducted on a group of university athletes and recreational runners, the amount of α-TNF decreased after vitamin D consumption [21], while other studies have found a negative correlation between 25 (OH)D concentration and TNF-α levels during a 24 h recovery period [34,36]. In addition, the levels of inflammatory markers in soccer players were investigated [39]. A vitamin D supplement with a daily dose of 3000 international units was used for 12 weeks. In this research, the results indicated that 12 weeks of vitamin D supplementation, despite optimizing vitamin D concentration, had no significant effect on IL-8, hs-CRP, and TNF-α levels in athletes. These findings regarding TNF-α levels are in agreement with the observations of Lewis [37], Zebroska [34], and Ikedo [36]. These results are inconsistent with the proposed association between vitamin D and inflammation in athletes and healthy adults [21]. Mieszkowski [38] recruited 35 ultramarathon runners to investigate the effect of vitamin D supplementation on inflammatory markers after long-term running. For this purpose, the runners were divided into two groups: the high-dose vitamin D supplement group and the placebo group. The supplement group took a dose of 150,000 international units of vitamin D 24 h before the run. Based on the post-test analysis, the increase in the level of interleukin 6 and 10 and resistin in the runners of the control group immediately after running was significantly higher than in the runners of the supplement group. IL-6 has a metabolic role as well as a regulatory role in inflammation. While IL-6 levels increase during exercise, it stimulates the mobilization of fatty acid metabolism as well as the inflammatory response. In addition, it has been shown that inhibition of IL-6 signaling improves inflammatory diseases [41]. In a study involving obese subjects, 25 (OH) D levels illustrated a significant inverse correlation with resting IL-6 levels [42]. Consistent with these results, and taking into account the changes in IL-6 levels immediately after the marathon in the vitamin D-supplemented group, a reduction in IL-6 levels 24 h after eccentric exercise in athletes treated with a low dose of vitamin D for 3 weeks (2 × 1000 units per day) was observed [34].

IL-10 is an anti-inflammatory cytokine whose serum level increases after exercise. Mieszkowski et al. [38] confirmed this in their study, as they observed that IL-10 levels increased significantly after exercise in the control group. Interestingly, this effect was not evident in the vitamin D-supplemented runners. Consistent with Matilainen [43], the IL10 gene contains regions that recruit the vitamin D receptor (VDR) in a ligand-dependent manner, and 1,25 (OH) D reduces IL10 expression. Furthermore, IL-6 release from active muscle is associated with increased circulating levels of the anti-inflammatory cytokine IL-10 [44]. Therefore, the decrease in IL-10 levels after running can also be explained by the decrease in the stimulation of IL-10 synthesis by IL-6. The authors also found no change in IL-15 levels after the intervention [38].

On the other hand, in the study by Kasprowicz [35], in which 20 male runners took part, the supplement group took a daily supplement of 10,000 IU of vitamin D for two weeks before implementing the intervention. The results of this research indicated that there was no significant difference in IL-6 after treatment with vitamin D.

Based on the existing literature, there are conflicting findings regarding the impact of vitamin D on pro-inflammatory cytokines in athletes. The varying doses of vitamin D, age range of participants, and intervention period duration are all factors that may have influenced the final results. As such, it is recommended that further studies be conducted with homogenized variables such as vitamin D dosage, gender, and sample size.

## 6. Conclusions

Our systematic review revealed that only two out of six studies demonstrated notable changes in IL-6 levels subsequent to the administration of vitamin D. Conversely, the remaining four studies did not exhibit significant alterations in interleukin-6 and TNF-alpha levels following vitamin D supplementation in athletes.

### Practical Implications

These consistent and contradictory results suggest that athletes are at a heightened risk of immune system deficiencies, underscoring the importance of further research in this area. It remains unclear as to whether vitamin D supplementation should be unequivocally recommended, necessitating further investigation in future studies. Consequently, this study may provide practicable information regarding the amount and duration of vitamin D intake for researchers wishing to conduct research in this area.

## Figures and Tables

**Figure 1 sports-12-00032-f001:**
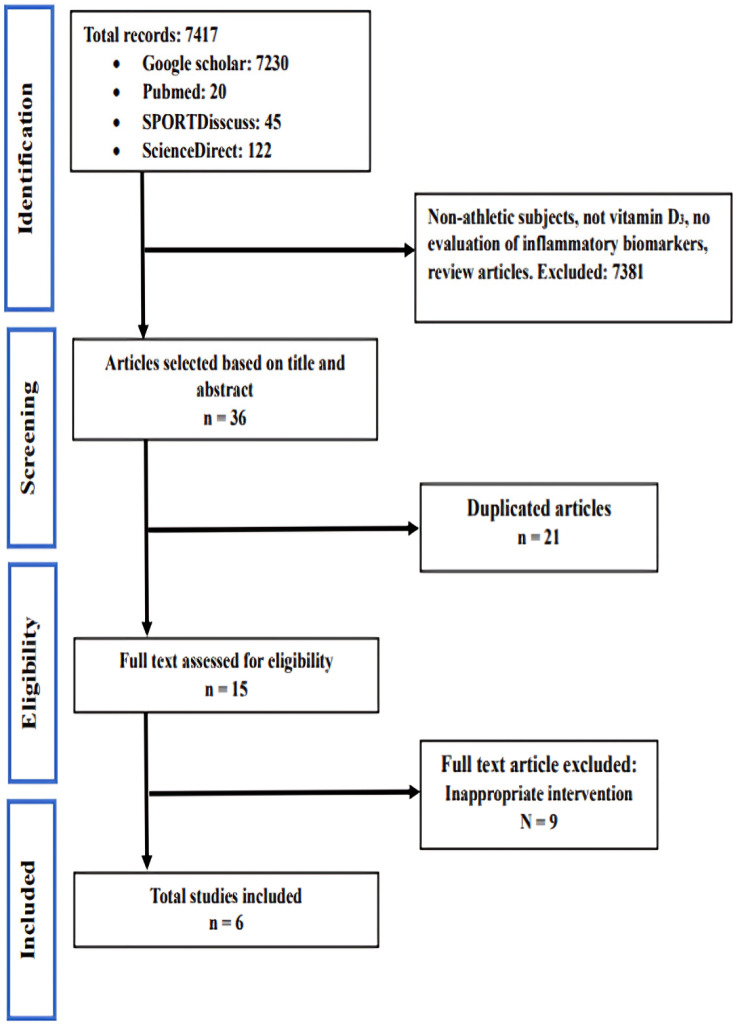
Selection of studies for inclusion. RCT: randomized controlled trial.

**Figure 2 sports-12-00032-f002:**
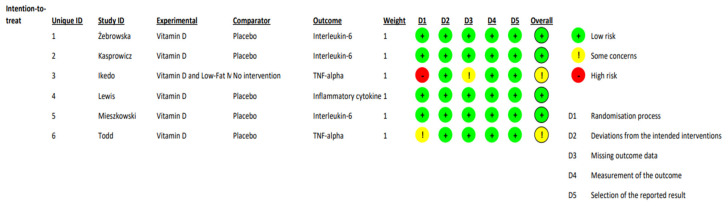
Risk of bias table.

**Table 1 sports-12-00032-t001:** Characteristics of the included studies.

Source, Year	Country	Sample Size	Participants	Sex	Groups	Age [Years]	Trial Duration	Vitamin D Situation	Vitamin D Consumption and Dosage	Exercise Protocol	Summary
Żebrowska et al. [34]	Switzerland	24	Ultramarathon Runner	M	EXP, n = 12 CON, n = 12	EXP: 33.7 ± 7.5 CON: 35.9 ± 5.3	3 W	None of the participants had a vitamin D deficiency or toxicity (<20 ng/mL or >100 ng/mL)	EXP: received 50 μg (2 × 1000 IU/day) of vitamin D; received a placebo (1.3 g lactose monohydrate)	Thirty-minute downhill running test with 70% VO_2peak_	Lower 1 h post-exercise TNF-α levels; a nonsignificant lower 1 h post-exercise IL-6 levels; a significant lower 24 h post-exercise IL-6 level
Kasprowicz et al. [35]	Poland	17	Ultramarathon Runner	M	EXP, n = 8 CON, n = 9	EXP: 39.0 ± 5.9 CON: 42.5 ± 8.3	2 W	None of the participants had a vitamin D deficiency or toxicity	EXP: group received 10,000 IU/day vitamin D	Run a 100 km distance	Non-significant changes in IL-6 level
J. J. Todd et al. [39]	Northern Ireland	42	Gaelic Footballers	M	EXP, n = 22 CON, n = 20	EXP: 20 ± 2 CON: 20 ± 2	12 W	None of the participants had a vitamin D deficiency or toxicity	EXP: group received an oral spray (3000 IU (75 μg))	Special football training	Non-significant changes in IL-8, hs-CRP, and TNF-α levels
Lewis et al. [37]	USA	32	Collegiate Swimmers and Divers	M/F	EXP, n = 19 CON, n = 13	EXP: 19 ± 1.6 CON: 19 ± 1.1	24 W	None of the participants had a vitamin D deficiency or toxicity	EXP: group received 4000 IU	Special swimming or diving training	Non-significant changes in TNF-α IL-6 and IL-1β
Mieszkowski et al. [38]	Poland	35	Ultramarathon Runner	M	EXP, n = 16 CON, n = 19	Nm	24 h	None of the participants had a vitamin D deficiency or toxicity	EXP: group were given a single high dose (150,000 IU) of vitamin D	Ultramarathon run	A significant decrease in IL-6 and 10 levels, but there was no effect on IL-15
Ikedo et al. [36]	Japan	26	High-school runners	F	EXP, n = 10 CON, n = 10	EXP: 16.3 ± 0.5 CON: 16.3 ± 0.6	24 W	None of the participants had a vitamin D deficiency or toxicity	EXP: consumed a vitamin D supplement (1000 IU/day) and low-fat milk (Ca 315 mg/day) for 6 months	Endurance Run	Non-significant changes inTNF-α

M: male; F: female; EXP: experimental; CON: control; W: week; h: hour; Nm: not mentioned.

## Data Availability

All data are available in the manuscript.

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
