# Peer review of "The Effect of Vitamin D Consumption on Pro-Inflammatory Cytokines in Athletes: A Systematic Review of Randomized Controlled Trials"

_sports, 2024, doi:10.3390/sports12010032_

Round 1

Reviewer 1 Report

Comments and Suggestions for Authors

Overall, the authors did a nice job compiling the related Vitamin D studies for review.

This review would have been stronger if it centered only on healthy individuals and dod not make any conclusions or leaps regarding "disease prone community" Athletes are not disease prone, the incidence of disease in athletics is similar to that of regular society. If anything, being athletic and involved in physical fitness enhances overall health and reduces mortality.

Including obese individuals or data from obese folk may also skew results as obesity itself has its own impact on circulating markers of inflammation (i.e., TNF-a, IL-6, IL-10, etc).

How do you better explain that varying doses of Vitamin D in the studies you analyzed - for how you controlled for such in making the overall assessment that you did? Certainly being intervened with 10,000 IU vs 2,000 IU of Vit D has different physiological impacts.

Section 5, line 201 "Cytokines that are focused on sport..." - please update your writing. A cytokine does not focus on anything, it has no brain. The start of line 208 is written wrong, no need to cite the year and author twice, please make more concise and uniform throughout the manuscript. This issue is true throughout the paper, please fix it no need to put (year) and then repeat it with (author, year), just use "author, year) throughout the document.

The paper would be stronger if listed  by dose studied and made somehow conclusions tied to dose and duration.

Author Response

Please see the attachment with the responses point-by-point.

Reviewer 2 Report

Comments and Suggestions for Authors

Manuscript titled “The effect of vitamin D consumption on pro-inflammatory cytokines in athletes: a systematic review of randomized controlled trials” is generally both the methodology and its structure are strong.

General comments:

There is inconsistency throughout the text in a way that states that vitamin D supplementation has no effect on IL-6 and TNF-α, while it appears that one or two (??) studies showed changes. Please correct this and be consistent.

References are not according to the MDPI journals rules, please correct.

Detailed comments:

Abstract

Lines 22–23: You are repeating that 6 studies included 173 subjects; please remove one.

First you mention that vitamin D supplementation has no effect on IL-6 and TNF-α, and after that, two studies reported significant changes in IL-6 and TNF-α levels after taking high-dose vitamin D. Please keep the same reporting, or you can state that studies reported opposite and inconclusive findings.

Also, introduce what the abbreviations IL-6 and TNF-α stand for in the abstract as well.

Line 67: „It reduces“ is redundant, and the sentence is generally not clear.

Methodology is well written.

Discussion

Lines 198-200: Again, you can not conclude that vitamin D supplementation does not affect IL-6 and TNF-α levels.

Line 231: Reference Mischowski is doubled.

Conclusion

Line 267: Only one? In the abstract, two studies are mentioned that showed changes??

Conclusion is generally very concise. However, additional information regarding future directions and practical implications are missing.

Comments on the Quality of English Language

Some tenses and grammatical errors (missing commas or full stops) are present throughout the manuscript. Please check carefully while revising the manuscript.

Author Response

(The authors gave the same response as above.)

Reviewer 3 Report

Comments and Suggestions for Authors

I thank the authors for the work done. The study is very interesting however I have some suggestions to give to the authors.

-       The abstract is well written and clear

-       The keywords should be different from the words in the title. This would increase the visibility of the indexed manuscript

-       The introduction of the functions or effects of vitamin D or its deficiency is quite good. However, I would like to suggest that the authors also discuss the role of vitamin D on the protection of muscle atrophy. Chronic inflammation, vitamin deficiencies, and an imbalance in the muscle-gut axis are just a few of the factors that can lead to sarcopenia.

(Agostini D, et al. An Integrated Approach to Skeletal Muscle Health in Aging. Nutrients. 2023 Apr 7;15(8):1802. doi: 10.3390/nu15081802. PMID: 37111021; PMCID: PMC10141535.)

-       Authors should use the PRISMA 2020. (The PRISMA 2020 statement: an updated guideline for systematic reporting reviews. BMJ. 2021;372:n71.)

-       How did you record the manuscripts? Did you use software or a simple Excel sheet? How were inclusions or exclusions assessed? Discussion between researchers? The description is unclear and not very precise

-       Consider whether to include “The Grading of Recommendations Assessment, Development, and Evaluation (GRADE)", framework to assess the certainty of the evidence in the systematic review. (Corbett MS, Higgins JP, Woolacott NF. Assessing baseline imbalance in randomised trials: implications for the Cochrane risk of bias tool. Res Synth Methods. 2014;5(1):79–85).

It would provide clarity and any limitations of the studies could be assessed. In 2023, Patti et al used in their review the GRADE profiler (GRADEpro GDT) to create a summary of the findings table. I would recommend using it.

-       Inserting the type of sport practiced by the samples in the table would give the reader greater clarity and ease of finding information

-       The age of the analyzed samples is very different in the studies taken into consideration. Could this be a limit?

-       The limitations of the study do not seem to be discussed in the discussion. They should be included

-       The discussion is quite clear; however, it is my opinion that it is a simple repetition of the results of the studies taken into consideration. The explanation of the results with the support of the literature and the opinion of the researchers after analyzing the data could be fundamental

Author Response

(The authors gave the same response as above.)

Round 2

Reviewer 3 Report

Comments and Suggestions for Authors

The manuscript is clear and very interesting. The authors followed the suggestions I gave. I have nothing else to recommend